# The Kinesin-12 Kif15 is a processive track-switching tetramer

**Hauke Drechsler[1], Toni McHugh[2], Martin R Singleton[3], Nicholas J Carter[1], Andrew D McAinsh[1]\***

[1]Mechanochemical Cell Biology Building, Division of Biomedical Cell Biology, Warwick Medical School, University of Warwick, Coventry, United Kingdom; [2]Systems Biology Doctoral Training Centre, University of Warwick, Coventry, United Kingdom; [3]Cancer Research UK, London Research Institute, London, United Kingdom

**Abstract** Kinesin-12 motors are a little studied branch of the kinesin superfamily with the human protein (Kif15) implicated in spindle mechanics and chromosome movement. In this study, we reconstitute full-length hKif15 and its microtubule-targeting factor hTpx2 in vitro to gain insight into the motors mode of operation. We reveal that hKif15 is a plus-end-directed processive homotetramer that can step against loads of up to 3.5 pN. We further show that hKif15 is the first kinesin that effectively switches microtubule tracks at intersections, enabling it to navigate microtubule networks, such as the spindle. hKif15 tetramers are also capable of cross-linking microtubules, but unexpectedly, this does not depend on hTpx2. Instead, we find that hTpx2 inhibits hKif15 stepping when microtubule-bound. Our data reveal that hKif15 is a second tetrameric spindle motor in addition to the kinesin-5 Eg5 and provides insight into the mechanisms by which hKif15 and its inhibitor hTpx2 modulate spindle microtubule architecture.

**\*For correspondence:**
A.D.McAinsh@warwick.ac.uk

**Competing interests:** The authors declare that no competing interests exist.

**Reviewing editor**: Tony Hyman, Max Planck Institute of Molecular Cell Biology and Genetics, Germany

## Introduction

The human mitotic spindle is a microtubule-based protein machine with bipolar geometry that mediates chromosome segregation (*Dumont and Mitchison, 2009*). Initial assembly of the spindle requires the separation of centrosomes by the homotetrameric plus-end-directed molecular motor Kif11 (Eg5)—a member of the kinesin-5 family (*Tanenbaum and Medema, 2010*). In vitro reconstitution experiments reveal that Kif11 can drive the outward sliding of anti-parallel microtubules thus providing a mechanistic basis for centrosome separation (*Kapitein et al., 2005*). Although essential for centrosome separation, Kif11 is not required to maintain subsequent spindle bipolarity due to the compensatory activity of the Kinesin-12 Kif15 (hKlp2) (*Tanenbaum et al., 2009*; *Vanneste et al., 2009*). Furthermore, overexpression of hKif15 can overcome the absolute requirement for hKif11 in centrosome separation (*Tanenbaum et al., 2009*; *Sturgill and Ohi, 2013*). This functional redundancy has led to a model in which hKif15, like Kif11, can slide apart anti-parallel microtubules. Kinesin-12 proteins from *Xenopus (xKlp2)*(*Wittmann et al., 1998*), sea urchin (*Rogers et al., 2000*), and rat (*Liu et al., 2010*) are dimeric plus-end-directed motors that are targeted to the spindle by the microtubule-associated protein (MAP) Tpx2 (targeting protein for xKlp2) (*Wittmann et al., 1998*). hKif15 is also recruited to the spindle microtubules by hTpx2 in human cells (*Tanenbaum et al., 2009*; *Vanneste et al., 2009*) with the interaction between dimeric hKif15 and hTpx2 hypothesised to enable the motor to cross-link and slide anti-parallel microtubules (*Tanenbaum et al., 2009*). However, this model has been challenged by recent cell-biological studies showing that hKif15 localizes to kinetochore (k)-fibres (parallel microtubule bundles) and contributes to the generation of forces that counter those generated by hKif11 (*Sturgill and Ohi, 2013*; *Vladimirou et al., 2013*). Understanding how hKif11 and hKif15 cooperate during mitosis has important implications for cancer therapy because the clinical efficacy of hKif11 inhibitors, currently used as

**eLife digest** Before a cell can divide, it produces an extra copy of all its chromosomes, and it must then ensure that each daughter cell ends up with one copy of each chromosome. During the division process, a structure called the spindle forms in the cell. This spindle is made of thread-like extensions called microtubules that grow from two poles at opposite ends of the cell. These microtubules are responsible for getting the chromosomes to line up in the middle of the cell, and then pulling half of the chromosomes to one end of the cell, and half to the other end. The cell then divides into two daughter cells.

Two motor proteins—so-called because they consume chemical energy to 'walk' along the microtubules—have important roles in this process: Kif11 motor proteins mainly drive the formation of the spindle and thus division of the chromosomes. A cell that does not contain Kif11 can only divide if it contains extra copies of a second motor protein called Kif15: this suggests that Kif15 can serve as some sort of back up for Kif11.

Normal cells only divide when new cells are needed for growth or to replace old cells that have died. Cancer cells, on the other hand, divide in a way that is not controlled. Drugs that interfere with Kif11 have been developed in the hope that they will stop cancer cells dividing, but these drugs have not been very effective in clinical tests, possibly due to the Kif15 back up. Scientists hope, therefore, that a better understanding of the role of Kif15 may lead to improved cancer treatments.

Drechsler et al. have isolated individual Kif15 motor proteins and used advanced microscopy techniques to study them in action. These experiments showed that Kif15 motor proteins can travel long distances along a single microtubule, and can also switch to a different microtubule at intersections. This movement of Kif15 is stopped when they bump into Tpx2 proteins, which are sitting on the microtubules. Together, these proteins can also form links between microtubules that can withstand high forces. These properties provide a starting point to understand how Kif15 can act as a back up for Kif11 in cells. In the future, it will be important to work out how Kif11 and Kif15 motor proteins work together in teams to build the spindle.

monotherapy, has proven largely disappointing (*Rath and Kozielski, 2012*). hKif15 is therefore emerging as a potentially important therapeutic target (*Groen, 2013*). To understand how hKif15/hKif11 operate within the spindle, we will require a detailed mechanistic understanding of how each motor interacts with microtubules and generates force. Such information is already available for Kif11. In this study, we provide the first insight into the properties of the Kinesin-12 family motor hKif15.

## Results

### hKif15 is a processive tetrameric motor protein

Full-length human His$_6$-Kif15 (hKif15), His$_6$-Kif15-eGFP, and His$_6$-Tpx2 (hTpx2) were expressed in insect SF9-cells and the recombinant proteins purified to near homogeneity by sequential affinity chromatography using a cation-exchange and a Co-NTA matrix (*Figure 1A*). Analysis of His$_6$-hKif15 on native PAGE revealed that the protein migrates at ~730 KDa (*Figure 1B*). Given the predicted molecular weight of His$_6$-hKif15 (164.8 kDa) our data suggest the presence of hKif15 tetramers, although hydrodynamic analysis of the frog (xKlp2) and Sea urchin (KRP$_{180}$) Kif15 orthologues indicated that the motor is dimeric with a sedimentation coefficient of 8.1S and 8.3S respectively (*Wittmann et al., 1998*; *Rogers et al., 2000*). To further characterise human Kif15, we subjected His$_6$-hKif15 to glycerol gradient ultracentrifugation on 5–40% glycerol gradients. At physiological ionic strengths (35 mM sodium phosphate buffer, 0-150 mM NaCl) both His$_6$-hKif15 and His$_6$-hKif15-eGFP appear to be mono-dispersed and have apparent sedimentation coefficients of ~12S (*Figure 1C*). However, increasing the salt concentration to 300 mM NaCl converts hKif15 into a species that runs at ~8S (*Figure 1C*). Taken together, our data show that hKif15 exists as a tetramer at physiological ionic strength that can be forced to dissociate into a dimer at high ionic strength. To further confirm that hKif15 can form tetramers we performed a size-exclusion chromatography combined with multi angle light scattering (SEC-MALS), which allows determination of the absolute molecular weight. This analysis confirmed the presence of

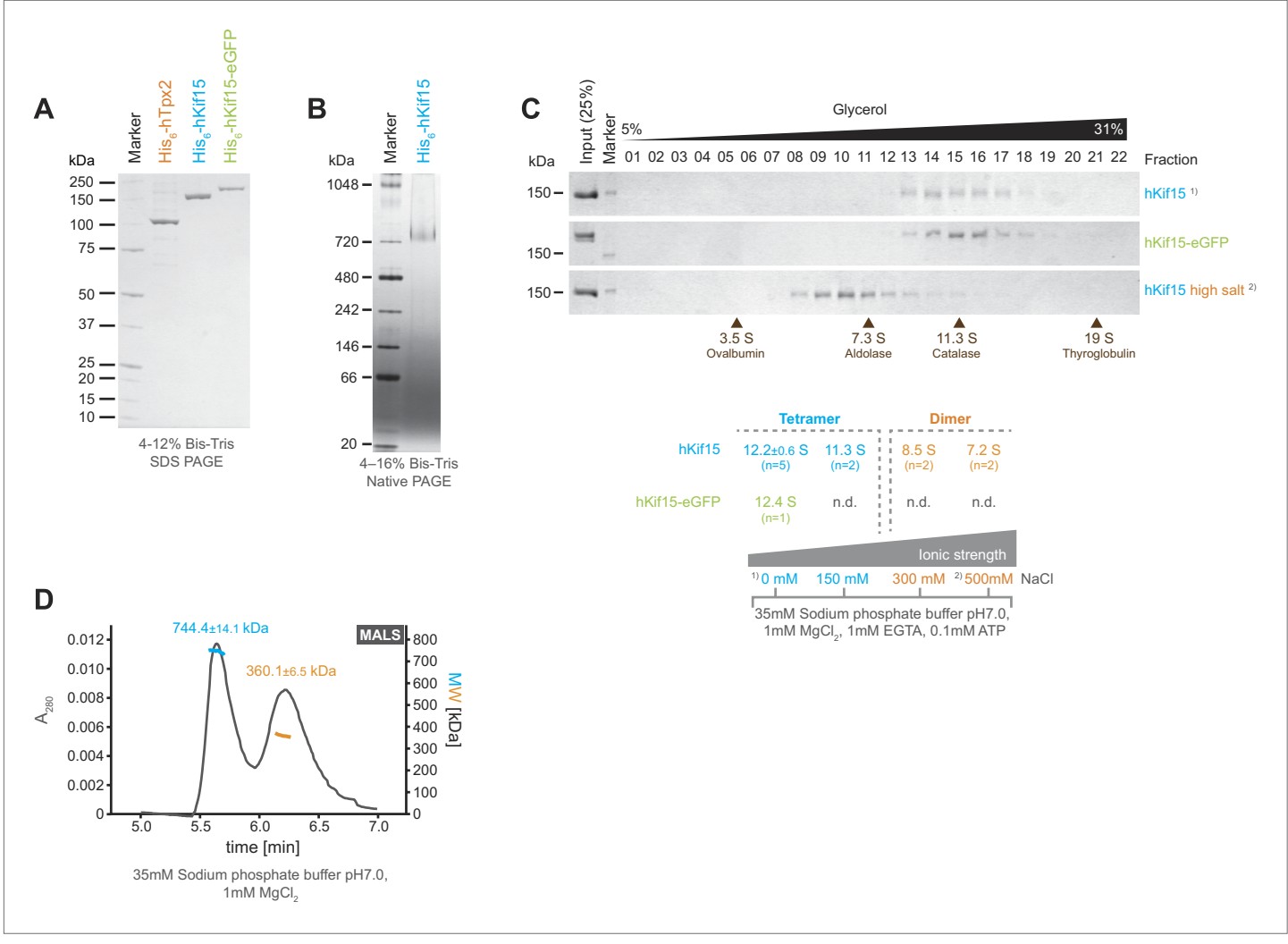

**Figure 1**. Kif15 is a tetramer. (**A**) Coomassie stained SDS-PAGE gel of purified His$_6$-hTpx2, His$_6$-hKif15, and His$_6$-hKif15-GFP. (**B**) Tetrameric His$_6$-hKif15 on a 4–16% Native-PAGE gel stained with coomassie. Calculated molecular weight: His$_6$-hKif15: 165 kDa. (**C**) Coomassie stained SDS-PAGE gels of fractions 1–23 out of 25 from 5–40% glycerol gradients loaded with either ~5 µg His$_6$-hKif15 or His$_6$-hKif15-eGFP at different salt concentration (see table 'below' that summarises the apparent S values of hKif15(eGFP) at different salt concentrations). (**D**) Elution profile (*grey line*, A$_{280}$, left y-axis) from a size-exclusion chromatography (SEC) run with subsequent multi angle light scattering (MALS) analysis. Outcome of the MALS analysis for the peaks is presented in coloured lines (*blue-hKif15 tetramer, orange hKif15 dimer*, MW, right y-axis). Molecular weight ± uncertainty is given above each peak. Please note that the presence of dimeric hKif15 species is specific to the protein preparation used in this experiment only ('Materials and methods'). Standard preparations do not contain any significant proportion of dimeric hKif15 (see panels **B** and **C** this Figure).

tetrameric hKif15 motors (molecular weight of 744.4 ± 14.1 kDa) (*Figure 1D*, *1st peak, blue line*). We could also detect a dimeric population of motors (molecular weight of 360.1 ± 6.5 kDa) indicating that there is some degree of complex dissociation in solution under these conditions (*Figure 1D*, *2nd peak, orange line*).

Next, we used total internal reflection fluorescence (TIRF)-microscopy, to observe the movement of single hKif15-eGFP motors on polarity-marked microtubules that were stabilised with the non-hydrolysable GTP-analogue GMP-CPP. These experiments were done in buffer conditions (BRB20, 50 mM KCl) that would be expected to preserve hKif15 in a tetrameric state (from here on hKif15-eGFP refers to single tetrameric motors). The majority hKif15 motors move either by directional movement towards the plus-end of individual microtubules (*Figure 2A*, '*P*') or by bidirectional 1D-diffusion (*Figure 2A*, '*D*', *Figure 2B*, *right*, *Figure 2—figure supplement 1D*, *right*) along the microtubule lattice. These types of movement are equally likely at all salt conditions tested (*Figure 2—figure supplement 1A*)

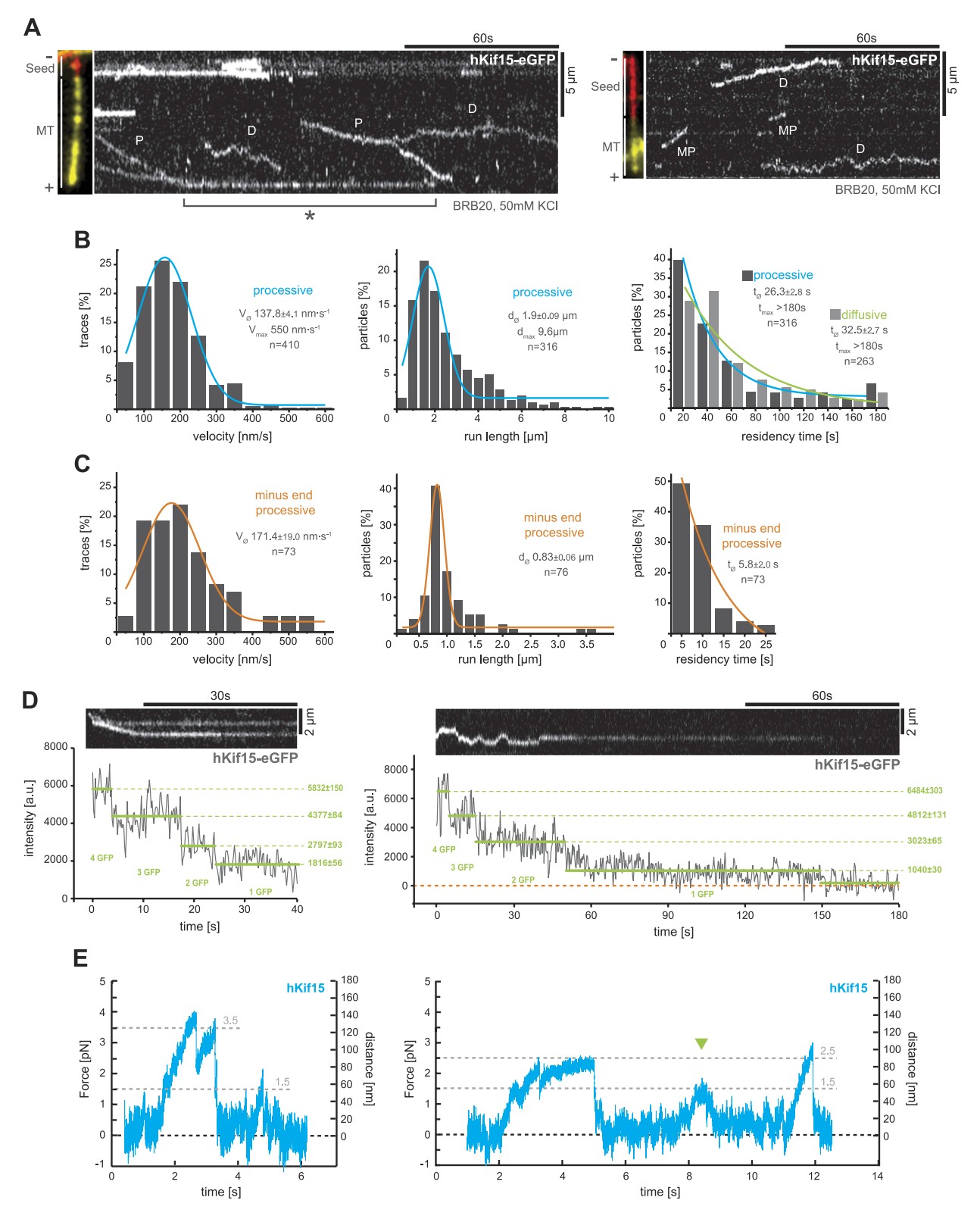

**Figure 2**. hKif15 is a processive tetramer. (**A**) Kymographs showing typical behaviour of eGFP-labelled hKif15 motors on GMP-CPP stabilised microtubules (*processive movement–P, diffusion–D, plus-end dwelling–asterix, processive minus-end directed movement–MP*). Pictures on the *left* of each kymograph show orientation of the polarity labelled microtubule. (**B**) Frequency distributions showing kinetic properties of processively moving plus-end-directed

*Figure 2. Continued on next page*

*Figure 2. Continued*

eGFP-labelled hKif15 motors. Coloured lines within the column plots are Gaussian or exponential decay fits. Insets show the respective median ± standard error of mean and maximum value of the distribution as well as its sample size. All values are derived from kymographs as shown in (**A**). (**C**) Frequency distributions showing kinetic properties of eGFP-labelled hKif15 motors that move processively to the minus-end of a microtubule. Coloured lines within the column plots are Gaussian or exponential decay fits. Insets show the respective median ± standard error of mean and maximum value of the distribution as well as its sample size. All values are derived from kymographs as shown in (**A**). (**D**) Kymographs showing a processively moving hKif15-eGFP motor (*left*) and a diffusive hKif15-eGFP motor (*right*) that photo-bleach in three and four steps respectively, indicating presence of four eGFP molecules in each motor. Plot below the kymograph shows the intensity along motor traces in arbitrary units. Intensity had been locally corrected for background intensity. Horizontal green lines within the graphs indicate the fitted average intensity (arbitrary units ± standard deviation) in the respective section of the trace. Photo-bleaching steps were manually defined. Please note that for the last bleaching step in the *right* kymograph we cannot formally exclude a dissociation event. (**E**) Example traces showing the movement of a single molecule (hKif15 tetramer) as a function of time (1 ms boxcar filtered). The motor steps out of the trap centre (*black dotted line*) until it detaches at variable loads between 1.5 and 3.5 pN and reattaches (in the trap centre) for subsequent movements. Movements can also be bidirectional (*green arrowhead*, compare with *Figure 2A*).

The following figure supplements are available for figure 2:

**Figure supplement 1**. TIRF-based analysis of hKif15 motility.

**Figure supplement 2**. Analysis of bead motility and microtubule-binding in laser trap experiments.

and single motors can switch between both modes (*Figure 2—figure supplement 1B*). Processively moving, plus-end-directed motors move at a speed of 137.8 ± 4.1 nm•s$^{-1}$ median ± SEM, though we observed velocities up to 550 nm•s$^{-1}$ (*Figure 2B*, *left*). The run length of hKif15 motors is 1.9 ± 0.09 μm median ± SEM (*max.* 9.6 μm) (*Figure 2B*, *middle*) and their residency time 26.3 ± 2.8 s median ± SEM (*max.* >180 s) (*Figure 2B*, *right*). Processive runs can include pauses up to 20.5 s (median ± SEM: 5.0 ± 0.4 s) at a frequency of 0.30 ± 0.04 pauses•μm$^{-1}$ mean ± SD (n = 162, three different experiments) without motor-dissociation from the microtubule lattice (*Figure 2—figure supplement 1B,D*, *middle*). Additionally, when motors reach the microtubule end, more than 90% remain attached (*Figure 2—figure supplement 1D*, *left* note orange column). Motors dwell at the tip for 18.8 ± 3.5 s median ± SEM (*max.* >152 s) (*Figure 2A*, (asterix), *Figure 2—figure supplement 1D*, *left*) and may switch to diffusional movement on the end-proximal microtubule lattice (*Figure 2—figure supplement 1C*). While the majority of hKif15 motors either moves directionally to the plus-end or diffuses along the lattice, we also observed a small proportion of motors that move processively towards the minus-end of polarity-marked microtubules (87.5% plus-end-directed motors vs 12.5% minus-end-directed motors, n = 504) (*Figure 2A*, *right kymograph, 'MP'*). Minus-end-directed motors move at a speed similar to plus-end-directed motors, but have a significantly reduced run length/residency time and do not undergo directional reversals (*Figure 2C*).

To confirm that motile hKif15 motors are tetramers, we imaged hKif15-eGFP motors under conditions that caused photobleaching and measured the change in fluorescence intensity with time ('Materials and methods'). This analysis showed that both processive and diffusive motors bleach in up to four steps, indicating the presence of four eGFP molecules per motor (*Figure 2D*). Furthermore, processive tetramers are evident throughout the velocity distribution (*Figure 2—figure supplement 1E*). These data are in line with our biochemical analysis (*Figure 1*) and confirm that the observed hKif15-eGFP motors are indeed tetramers.

Within the mitotic spindle, motors will experience varying loads and thus it is essential to understand how a motor responds to force. We therefore investigated the mechanical properties of hKif15 motors using our single-bead laser trap set-up (*Carter and Cross, 2005*). Single hKif15 motors were adsorbed onto 550-nm diameter polystyrene beads and steered to the microtubule surface with the laser trap at which point the motor can bind to the lattice (see 'Materials and methods' for details and *Figure 2—figure supplement 2A*). With the trap turned off (zero-load) the mean velocity of beads moving on the microtubule was consistent with our measurements of hKif15-eGFP by TIRF microscopy (*Figure 2—figure supplement 2B*). Next, we held the bead in the trap and allowed the motor to walk along the microtubule, away from the centre of the trap. The motion of the bead reports the stepping actions of the motor. *Figure 2E* shows two position traces in which hKif15 motors make a number of processive steps along the microtubule before detaching at relatively low loads between 1.5 and 3.5 pN. We also observed that beads move out the trap before moving back in the opposite direction without detaching

(*green arrowhead*). These events may reflect the minus- and plus-end-directed diffusive motion observed in our TIRF experiments (*Figure 2A*). Together with our TIRF microscopy, we conclude that hKif15 is a homotetramer that undergoes either diffusive movement along microtubules, processive plus-end-directed stepping over long-distances or short duration movements to the minus-end.

## hKif15 effectively switches microtubule tracks

Our single molecule TIRF experiments also revealed a remarkable property of hKif15 motors as they approach microtubule intersections. The well-studied kinesin-1 will normally pass intersections and only in rare cases the motor will pause or switch microtubule tracks (*Ross et al., 2008*). In contrast, hKif15 motors show significantly elevated pause and switch events at intersections. Every fifth motor that approaches an intersection switches its microtubule track (*Figure 3A,B*; *Videos 1 and 2*). In general all events—pass, switch, pause, and dissociation at intersections—are equally likely. In this way, hKif15 is able to roam microtubule networks and *Figure 3C* demonstrates that one tetramer (see *bleaching steps, lower panel*) can indeed navigate along several different microtubules: the motor starts processive movement on microtubule #1 (trace A), eventually dissociates and rebinds to microtubule #2. On this, the motor diffuses a short distance (trace B) and dissociates again, just to rebind to microtubule #1. It processively follows microtubule #1 (trace C) and sequentially switches to microtubules #3 (trace D) and #4 (trace E), ending at the plus-end of microtubule #4 without detaching from the microtubule lattice(s) once again (see also *Video 3*). Thus, Kif15 is a processive motor that is able to navigate microtubule networks by effectively switching microtubule tracks.

## hKif15 confers dynamic microtubule crosslinks

The tetrameric organisation of hKif15 would support the hypothesis that the motor drives extensile sliding of anti-parallel microtubule overlaps like the tetrameric Kif11 motor (*Tanenbaum et al., 2009*). To establish, whether tetrameric hKif15 indeed is able to cross-link microtubules on its own, we carried out experiments, in which biotinylated HiLyte-647-labelled GMP-CPP-microtubules were attached to a cover slip followed by addition of short XRhodamin-labelled GMP-CPP-stabilized- 'transport' microtubules and 1.4 nM tetrameric hKif15-eGFP. By drawing kymographs, we found that hKif15 could mediate the cross-linking of microtubules (*Figure 4*) on its own, but that there was very little microtubule–microtubule sliding activity. Besides pivoting microtubules, we did observe episodes of bidirectional short duration sliding indicating a tug-of-war (*Figure 4*, *right*) and rarely the transport of short microtubules (*Figure 4*, *left*): behaviours that were recently reported for the yeast Kinesin-8 Kip3 (*Su et al., 2013*). However, we have no clear evidence to support the idea that hKif15 can drive continuous extensile sliding of anti-parallel microtubule bundles like hKif11. This is presumably because the motors frequently pause during processive excursions on the lattice and dwell at the ends for a long period (*Figure 2—figure supplement 1B–D*). Consistent with this, we only observed discontinuous short distance movement and pivoting of microtubules (around their ends) on surface-bound hKif15 motors, but not continuous microtubule sliding in microtubule gliding assays (*Videos 4 and 5*).

## hTpx2 selectively inhibits hKif15 stepping

So far our experiments show that hTpx2 is not required for hKif15 to bind microtubules or for the motors processivity and ability to crosslink microtubules (as a tetramer). However, cell-biological experiments showed that hTpx2 is essential for hKif15 to be recruited to the mitotic spindle (*Tanenbaum et al., 2009*; *Vanneste et al., 2009*). To resolve this contradiction, we first analysed possible complex formation of recombinant hKif15 and hTpx2 in solution by ultracentrifugation in glycerol gradients. With nanomolar concentrations of each protein, we could not observe any complex formation over a range of different salt conditions (50 mM HEPES pH7.5, 0/150/300/450 mM NaCl [data not shown] and 35 mM sodium phosphate buffer [*Figure 5A*]). However, a 30-fold excess of hTpx2 dimers (2.5 µM hTpx2 [dimer] + 80 nM hKif15 [tetramer]) allowed the formation of a hKif15-hTpx2 complex of ~14.0S, which included approximately one third of the provided hKif15 (*Figure 5A*, *lower gels*). The dissociation constant of the hKif15–hTpx2 complex is likely to be in the µM range arguing that its formation in vivo is less likely. In contrast, using a microtubule sedimentation assay, we observed that at low nM concentrations, hTpx2 (30 nM, dimer) does enhance the affinity of hKif15 (15 nM, tetramer) binding to GMP-CPP stabilised microtubules when in the presence of the non-hydrolysable ATP analogue AMP-PNP (the fitted $K_d$ increasing ~twofold from 993 nM to 490 nM; *Figure 5B*). This observation could explain why cell-biological experiments show that association of hKif15 with the mitotic spindle depends on

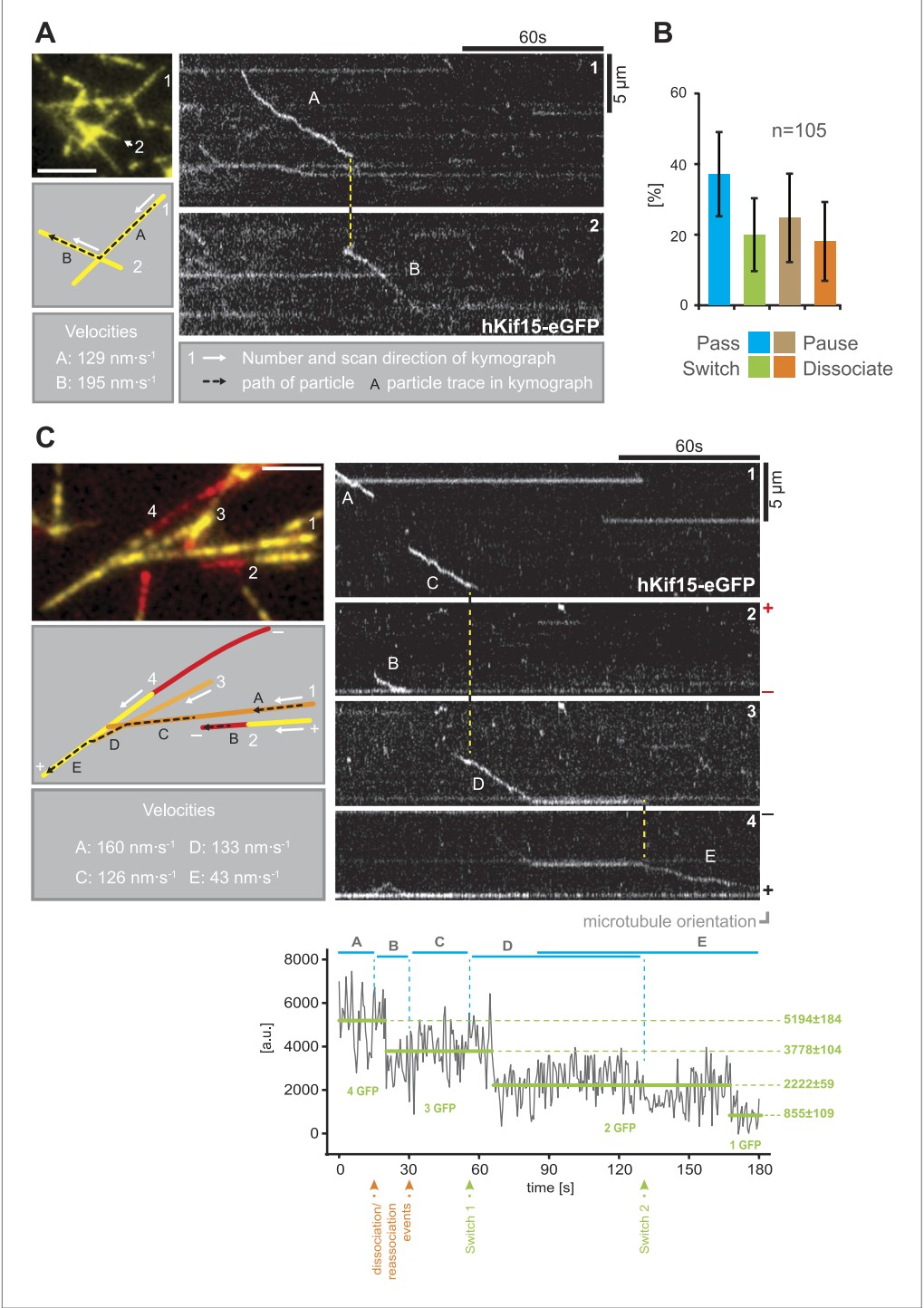

**Figure 3**. hKif15 effectively switches microtubules at intersections. (**A**) Kymographs showing the processive switch of an eGFP-labelled hKif15 motor at a microtubule–microtubule intersection. The image in the *upper left* gives an overview of the microtubule positons, *bar = 5 µm*. The schematic *below* depicts the microtubules whose line scans are depicted as kymographs on the *right*. The numbering of the microtubules corresponds to that of the respective kymograph and the assigned letters to the traces within the kymograph. White arrows depict the direction of the line-scan; the dotted black arrow depicts the path of the hKif15 motor. Velocities of single trace-sections are given. Please note that the motor never leaves the microtubule lattice during the switch event (yellow dotted line between the kymographs). (**B**) Quantification of motor behaviour at microtubule intersections. Definitions as following: *Pass*–motor passes on the same microtubule and continues movement (may include a pause at the intersection); *Switch*–motor switches the microtubule and continues
*Figure 3. Continued on next page*

*Figure 3. Continued*

movement (may include a pause at the intersection); *Pause*–motor pauses for more than 5 s and does not continue movement after intersection (may dissociate or bleach at some point); *Dissociate*–motor immediately dissociates at intersection. (**C**) Essentially as in (**A**), now following a motor along four microtubules. Movement involves two dissociation/re-association (Kymographs: *trace A/trace B*, *trace B/trace C*) and two microtubule switch events (*trace C/trace D*, *trace D/trace E*) during switch events the motor never leaves the microtubule lattice (yellow dotted line between the kymographs). Plot below the kymograph shows the intensity along motor traces in arbitrary units. Intensity had been locally corrected for background intensity. Horizontal green lines within the graphs indicate the fitted average intensity (arbitrary units ± standard deviation) in the respective section of the trace. Photo-bleaching steps were manually defined and show that the observed motor bleaches in three steps indicating presence of four eGFP molecules.

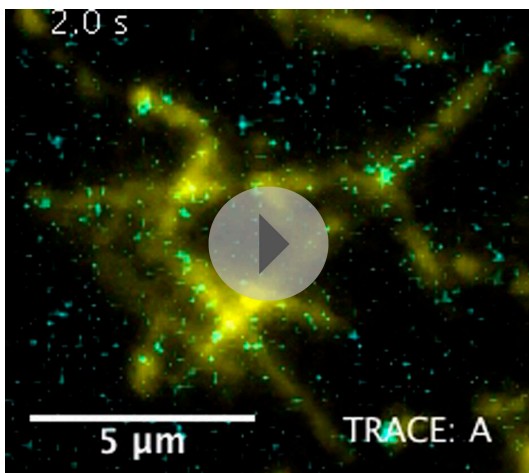

**Video 1**. hKif15 effectively switches microtubules at intersections. Video of the events summarised in *Figure 3A* (*hKif15-eGFP*—cyan, *microtubules*—yellow). The switching hKif15-eGFP motor is marked by the cyan arrowhead.

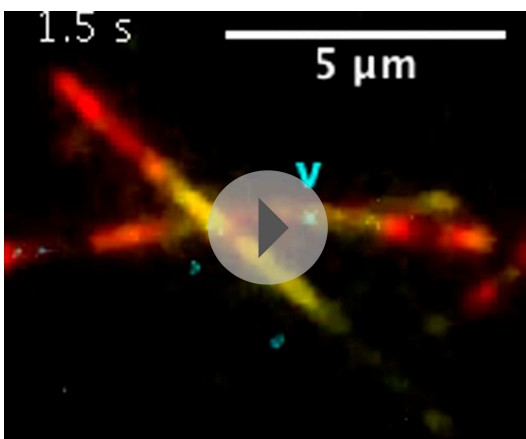

**Video 2**. hKif15 effectively switches microtubules at intersections. Video of another hKif15-eGFP motor switching polarity-marked microtubules (*hKif15-eGFP*—cyan, *seed*—red, *microtubule extension*—yellow). The switching motor is marked by the cyan arrowhead. Please note that the motor moves for some time towards the minus-end of the microtubule it switched onto and subsequently changes direction towards the plus-end.

hTpx2 (*Tanenbaum et al., 2009*; *Vanneste et al., 2009*). It is, however, unclear why hTpx2 further increases the microtubule binding affinity of a motor that is already highly processive.

To address the impact of hTpx2 on moving hKif15 tetramers, we added hTpx2 in equimolar (i.e., 5 nM) concentration to our hKif15-eGFP TIRF setup. Addition of hTpx2 decreased the proportion of motile motors on microtubules by 70%. The remaining motors still showed both directional and diffusive movement, indicating that inhibition by hTpx2 is not selective for either type of movement (*Figure 5C*, *kymographs and left graph below*). Further, the velocity distribution of remaining processive motors (compared to *Figure 2A*) now shows a bimodal distribution with a high proportion of slow motors (42.1 ± 1.7 nm•s⁻¹ median ± SEM) and fewer motors at the median speed of hKif15 motors in the absence of hTpx2 (114.9 ± 3.2 nm•s⁻¹ median ± SEM) (*Figure 5C*, *right graph below*). This may reflect the higher probability of a fast moving motor from engaging hTpx2 molecules that are bound on the microtubule lattice.

To further confirm the inhibition of hKif15 by hTpx2 and to estimate the functional impact of hTpx2 inhibition on hKif15 functions beside movement, we included hTpx2 in our optical trap experiment. To do this, we captured a hKif15-linked bead and confirmed that it underwent stepping on the microtubule (*Figures 5D*, *1. blue trace*). We then exchanged the buffer to allow 18 nM dimeric hTpx2 access to the hKif15-microtubule complex and again recorded the bead position. The hKif15 motor could no-longer step processively, although it remained bound to the microtubule lattice (4/5 cases; *Figures 5D*, *2. blue trace*). To rule out that the bead had not simply detached, we moved the microscope stage and observed an increase in force indicating an intact microtubule-motor connection (see silver asterisk in *Figure 5D*). During this period, the bead can maintain microtubule attachment at forces that—in the absence of hTpx2—would have forced dissociation of the

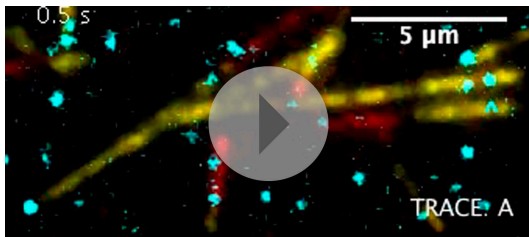

**Video 3**. hKif15 effectively switches microtubules at intersections. Video of the events summarised in *Figure 3C* (*hKif15-eGFP*—cyan, *seed*—red, *microtubule extension*—yellow). The switching hKif15-eGFP motor is marked by the cyan arrowhead.

motor (compare also with *Figure 2E*). Thus, hTpx2 increases the force-holding capability of microtubule-bound hKif15. We can rule out a generic 'roadblock' inhibition-mechanism: Firstly, kinesin-1 motility is unaffected by the presence of hTpx2 (n = 5; *Figure 5D*, *green traces*). Secondly, hTpx2 already inhibits hKif15 motility at low nanomolar concentrations when only single molecules are present on the lattice. Kinesin-1 motility in the presence of hTpx2 also shows that inhibition of bead motility in case of hKif15 is not simply mediated by hTpx2-bead-microtubule interactions. Thus, hTpx2 is a selective inhibitor of hKif15 motor stepping presumably by forming stable trimeric hKif15-hTpx2-microtubule complexes.

## Discussion

In this study, we reveal that hKif15 is a second tetrameric spindle motor in addition to the tetrameric Kinesin-5 hKif11 (Eg5). This finding is in line with initial cell-biological experiments that concluded that these two motors are redundant with Kif15 becoming essential for spindle assembly when hKif11 is inhibited (*Tanenbaum et al., 2009*; *Vanneste et al., 2009*). Indeed, hKif15, like Kif11, is only able to step under low loads in the 1–3 pN range (*Korneev et al., 2007*). However, our data show that Kif15 motors have a number of biophysical properties that distinguish it from Kif11: while both motors are able

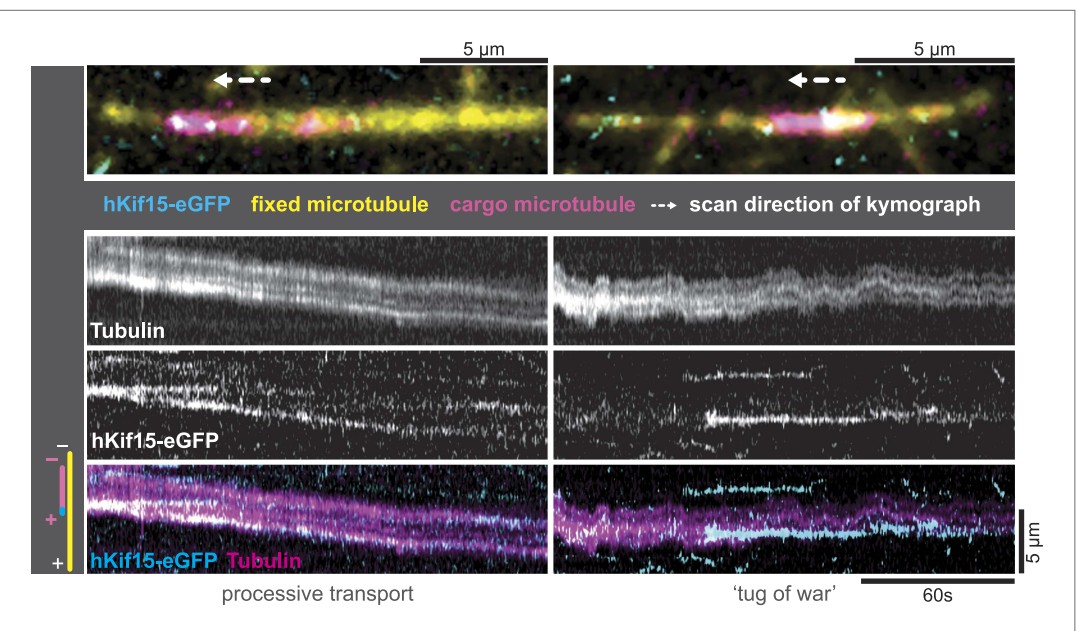

**Figure 4**. hKif15-eGFP can transport short microtubules as a cargo. Kymographs showing examples of processive (*left*) and tug-of-war-type transport (*right*), with an overview image of the crosslinked microtubules *on top*. Left: hKif15-eGFP motors drive slow (26 nm•s⁻¹) processive movement of a cargo microtubule. The moving microtubule (*magenta*) is attached via Kif15 motors (*cyan*) to a substrate microtubule (*yellow*) for which the plus-end is orientated left in the overview image on top and at the bottom of the kymograph. hKif15 motors can be seen stably associated with one end of the cargo microtubule, which must be the plus-end because the motors move in a plus-end direction and pause at the end (see *Figure 2A*, *Figure 2—figure supplement 1C*, see also schematics to the left). Thus, the cargo microtubule moves plus-end leading towards the plus-end of the substrate microtubule. That is, the microtubules are parallel. *Right*: hKif15-eGFP motors drive tug-of-war-type movement, characterised by frequent and rapid direction changes during movement.

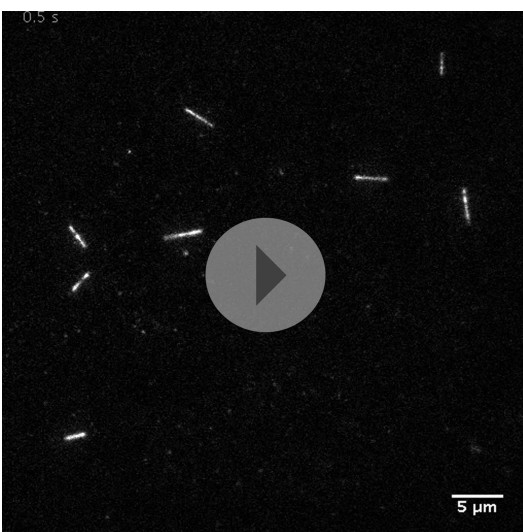

**Video 4**. Surface-bound hKif15 does not support continuous sliding of GMP-CPP stabilised microtubules (*white*) over long distances.

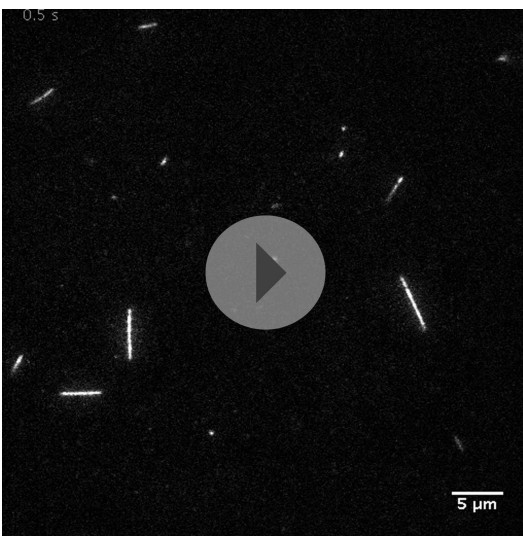

**Video 5**. Surface bound hKif15 does not support continuous sliding of GMP-CPP stabilised microtubules (*white*) over long distances.

to undergo plus-end-directed movement or 1-D diffusion (*Kwok et al., 2006*; *Kapitein et al., 2008*; *Figure 6*, 'A'), the median velocity of hKif15 is 10-fold higher than that of Kif11 and processive runs have a fourfold longer duration (*Korneev et al., 2007*). In the case of hKif15, both these movement types occur at the same ionic strength. In contrast, processive and diffusive movement by Kif11 is exclusive and depends on the ionic strength (*Kapitein et al., 2008*). Additionally, to our knowledge, pauses during processive runs have not been reported for Kif11.

The uniqueness of hKif15 is also reflected in the motors ability to track-switch during processive movement—a property shared with Dynein (*Ross et al., 2008*). It is easy to comprehend how dynein switches between microtubule tracks, as this motor is capable of making variable step sizes, significantly larger than 8 nm (*Reck-Peterson et al., 2006*). However, kinesins step with a step size of ~8 nm (*Svoboda et al., 1993*; *Valentine et al., 2006*; *Yardimci et al., 2008*; *Huckaba et al., 2011*; *Jannasch et al., 2013*) suggesting that it is unlikely for a single hKif15 motor domain-pair to mediate switching without leaving the lattice. In this regard, we show that within our 500 ms time resolution single motors do not leave the lattice during switching (*Figure 3*). Furthermore, switching events often include a pause at the intersection (*Figure 3C*, traces 'D' and 'E'). As hKif15 is a motile tetramer, we propose that a switch starts with attachment of the second, free head-pair to the lattice of the intersecting microtubule, while the first, engaged head-pair of the same molecule stays attached to the old track. This hKif15 bridge can now pause at the intersection or resolve by detachment of the second domain-pair into a pass or by detachment of the firstdomain-pair into a switch event (*Figure 6*, 'B'+box).

Microtubule switching is a feature that is very helpful to navigate through the complex microtubule arrangement within the k-fibre. Thereby each switch event is equivalent to a new independent run on a new microtubule, so that a single molecule can travel a distance of '*median run-length* × ($n_{switches}+1$)' as we have shown in *Figure 3C*: traces C-E sum up to 7.4 μm run length in total, with single traces at 1.6, 3.7 and 2.1 μm. Thus, even with a moderate median run length of 2 microns, a single molecule would have a high probability of reaching the plus-end (kinetochores) of the k-fibre, regardless of any discontinuity within the fibre or roadblocks owing to structural microtubule-associated elements (*Figure 6*, 'B'+box). In contrast, short running, non-switching hKif11, that additionally is constantly gathered at the spindle poles by a dynein-dependent minus-end-directed flux (*Ma et al., 2010*), cannot travel far enough into the k-fibre network to reach the kinetochore/plus-end. In vivo data backs up this hypothesis, as hKif11 localises mainly to the spindle pole and fades towards the spindle midzone, while hKif15 is localised uniformly along the k-fibres (*Vanneste et al., 2009*; *Sturgill and Ohi, 2013*). Moreover, Kif15 motors also modulate the capacity of k-fibre microtubule bundles to generate an inward-directed force within the spindle (in opposition to hKif11) (*Sturgill and Ohi, 2013*). One

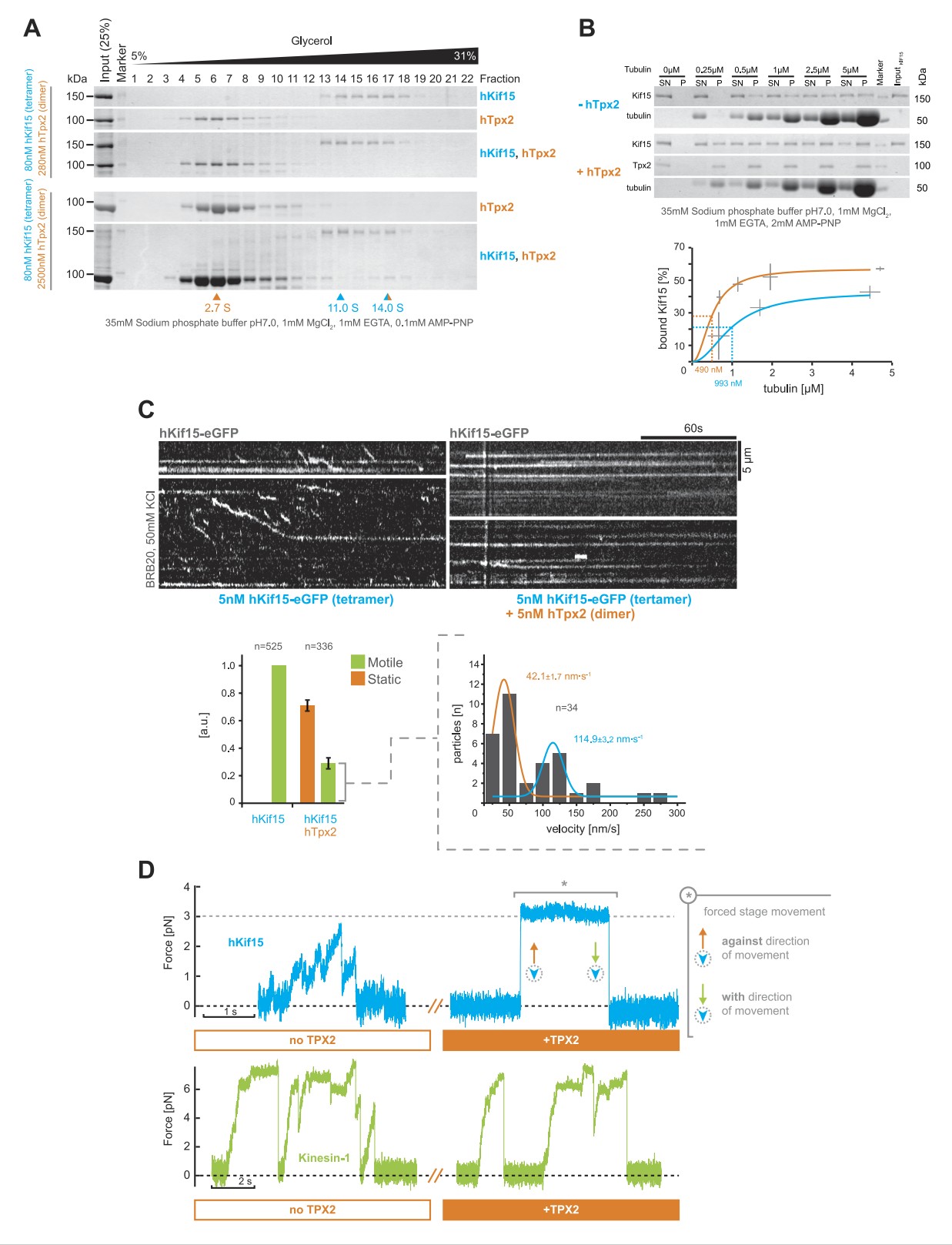

**Figure 5**. hTpx2 inhibits hKif15 motility by increasing its microtubule affinity. (**A**) Formation of a stable hKif15/hTpx2 complex occurs only at low μM concentration of Tpx2. Coomassie stained SDS-PAGE gels showing the first 23 of 25 fractions of a 5–40% glycerol gradient in 35 mM sodium phosphate buffer loaded with the indicated purified proteins alone and in mixture at the indicated concentrations. While hKif15 and hTpx2 do not form a stable

*Figure 5. Continued on next page*

*Figure 5. Continued*

complex at nanomolar concentrations, a vast excess of 2.5 μM dimeric hTpx2 drives formation of a hKif15/hTpx2 complex of 14.0S. (**B**) *Above*: coomassie stained SDS-PAGE gel of a typical microtubule co-sedimentation experiment of 15 nM tetrameric hKif15 in the presence or absence of 30 nM dimeric hTpx2 at different concentrations of taxol-stabilised microtubules (*SN–supernatant, P–pellet*). *Below*: Quantification of pelleted tubulin and bound hKif15 shown *above*. Crosses indicate the SD of the average from three independent experiments. Deviation in tubulin concentration is due to partial microtubule instability in phosphate buffer at low tubulin concentration and the microtubule stabilising effects of hTpx2 (compare pelleted tubulin at 0.25 μM with and without hTxp2). (**C**) Kymographs show the effect of 5 nM dimeric hTpx2 on the motility of 5 nM tetrameric hKif15 on GMP-CPP stabilised microtubules. Graph below *left* shows the fraction of motile and static motors normalised to overall microtubule length and subtracted by the fraction of static motors in the control, which sets motile motors in control to 1. Error bars show SD of three independent experiments. Graph below right shows the velocity distribution of motile motors in the presence of hTpx2, coloured lines are Gaussian fits revealing a bimodal distribution, compare with *Figure 2B*. (**D**) Side-by-side comparison of hKif15 (blue) and Drosophila kinesin-1 (green) stepping in the absence and presence of hTpx2 in the laser trap. Stepping traces for each kinesin are from the same bead before flow-in of 36 nM hTpx2 (*left trace*), after flow-in of hTpx2 (*right trace*). The asterix in the above *right trace* marks the deliberate movement of the stage, showing motor maintained attachment to the microtubule.

possibility is that, due to their prolonged end dwell time, hKif15 motors may influence plus-end dynamics at kinetochores (*Figure 6*, '*C*'). Such is the proposed role of the Kinesin-8 Kif18a, which is reported to dampen microtubule dynamics in vitro (*Du et al., 2010*; *Stumpff et al., 2012*).

Tetrameric hKif15 motors, like Kif11, can form cross-links between microtubules, we show that hKif15 crosslinks are dynamic as we observed limited tug-of-war movements of crosslinked cargo microtubules. Such cross-links may be reinforced by inhibitory hTpx2 (see below) turning dynamic

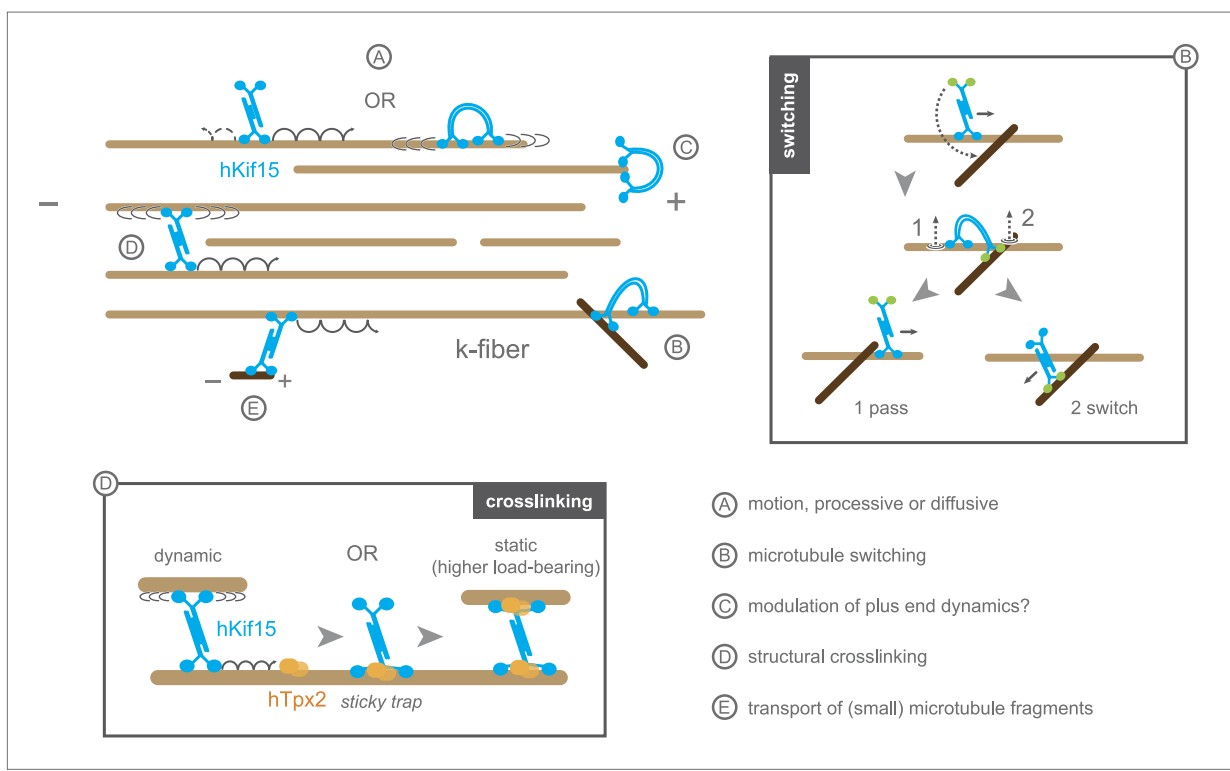

**Figure 6**. Schematic model summarising the biophysical and biochemical properties of hKif15 and illustrating how the motor may operate within a k-fibre (parallel microtubule bundle). *A*–hKif15 can move uni-directionally towards the plus-end (minus-end-directed motion can also occur albeit with lower frequency) by processive stepping or by bi-directionally diffusion along the microtubule lattice. *B*–hKif15 can switch between microtubule tracks. (*box*) Model for the sequence of events during switch or pass movements at intersections via a bridge structure of the hKif15 tetramer that resolves into a pass (1) or switch event (2) depending on which motor domain pair detaches first from the microtubule lattice. *C*–hKif15 has a significant plus-end dwell time and therefore might modulate plus-end dynamics like other k-fibre motors (*Stumpff et al., 2012*). *D*–Being a tetramer, hKif15 can crosslink two microtubules and potentially resist sliding of microtubules within the fibre. Note that this link is dynamic as the motor still can step or diffuse within a bundle. (*box*) Once hKif15 and hTpx2 have formed a complex on the microtubule lattice, this crosslink could become static, thereby forming a fixed structural link in the fibre, which can sustain higher loads than hKif15-only crosslinks. *E*–hKif15 powers transport of (small) microtubule fragments.

hKif15 crosslinks into static structural hKif15/hTpx2 crosslinks that can resist higher forces than hKif15 only crosslinks (*Figure 6*, '*D*'+*box*). Our data does not, however, provide any evidence that hKif15 drives a Kif11-like continuous extensile sliding of anti-parallel microtubules in vitro, though we cannot rule out that in vivo other additional factors are involved in such an activity. This factor is unlikely to be hTpx2 since our data shows this protein to inhibit the stepping of hKif15. How this mechanism is regulated within the spindle will be an important future work, but the fact that hTpx2 and hKif11 are constantly transported to the spindle pole (*Ma et al., 2010*) might hint to a partition of the spindle into a hKif11 active zone around the poles and a hKif15 active zone towards the spindle equator.

Overall, our data provide the first insight into the biochemical and biophysical properties of the full-length human Kinesin-12 hKif15. We reveal that hKif15 is a distinct class of kinesin that assembles into stable tetramers, which are highly processive, can navigate microtubule networks by switching track and form high load-bearing microtubule–microtubule crosslinks when bound to the regulatory factor hTpx2. These data shed new light on the mechanism by which hKif15 motors control spindle and chromosome mechanics.

## Material and methods

### hKif15 open reading frame cloning and expression plasmid construction

The hKIF15 ORF was amplified in five 800- to 1000-bp sized fragments from a cDNA library (derived from hTERT immortalised retinal pigment epithelial (hTERT-RPE1 cells)) and subsequently joined by overlap extension PCRs using Pfu Ultra AD polymerase (Agilent, Stockport, UK). The complete hKIF15 ORF was cloned via *Ase*I/*Not*I into a modified (bases 2391–2426 of the pIEx/Bac1-vector coding the Strep-Tag II were replaced by the His$_6$/TEV-cleavage site element of p11 [DNASU Plasmid ID: EvNO00085126, DNASU Plasmid Repository at Arizona State University], flanked by a 5′ *Nco*I and a 3′ *Nde*I site. [gift form Miho Katsuki, Riken, Japan]) pIEx/Bac1-vector (MERCK, Darmstadt, Germany) opened with *Nde*I/*Not*I. For hKIF15-eGFP an *Asc*I-site was introduced in front of the *Not*I-site and the eGFP inserted via *Asc*I/*Not*I before the hKIF15 ORF was inserted as above. We observed that the hKIF15 ORF is toxic to *Escherichia coli* cells when antibiotics other than ampicillin (e.g., kanamycin, gentamycin) are used, so viral genome generation by the Invitrogen Bac-to-Bac system (Life Technologies, Paisley, UK) using DH10BAC cells is not possible. The hTPX2 ORF was amplified from I.M.A.G.E. 3509275 (ATCC clone MG1537) and cloned into the pFastBacM13 vector (MPI-CBG, Dresden, Germany) via SpeI/SalI.

### Protein expression and purification

Assembly of viral genomes was carried out according to manufacturer protocols and transfection competent baculovirus particles were generated and used for transfection of 500 ml–1 L SF9-cells expression cultures according to (*Wasilko et al., 2009*). Cells were harvested at 250-g in a SLA-3000 rotor (Thermo Scientific, Waltham, MA, USA) and resuspended in lysis buffer (50 mM HEPES pH 7.5, 150 mM NaCl, 1.5 mM MgCl$_2$, 3 mM EGTA, 5% glycerol, 0.1% Tween-20, 0.1 mM ATP, complete protease inhibitor [Roche, Burgess Hill, UK]).

For hKif15 purification, lysates were cleared at 48k-g in a SS-34 rotor (Thermo Scientific), diluted 1:3 (to 50 mM NaCl final) with 50 mM sodium phosphate buffer and adjusted to pH 7.0. Protein was allowed to bind to SP-sepharose (GE Healthcare, Little Chalfont, UK) in batch for 2 hr at 4°C. hKif15 was directly eluted by applying a threefold bed volume of 50 mM sodium phosphate buffer pH 7.5, 50 mM NaCl, 1 mM MgCl$_2$, 5% glycerol, 0.05% Tween-20, 0.1 mM ATP onto Talon-beads (Takara Biotech, Saint-Germain-en-Laye, France). Again, protein was allowed to bind 2 hr at 4°C in batch. Talon-beads where washed with 10-fold bed volume of 50 mM sodium phosphate buffer pH 7.5, 300 mM NaCl, 1 mM MgCl$_2$, 5% glycerol, 0.05% Tween-20, 0.1 mM ATP and a 15-fold bed volume of 50 mM sodium phosphate buffer pH 7.5, 100 mM NaCl, 1 mM MgCl$_2$, 5% glycerol, 0.05% Tween-20, 10 mM imidazole, 0.1 mM ATP. Protein was eluted with 50 mM sodium phosphate buffer pH 7.5, 150 mM NaCl, 1 mM MgCl$_2$, 10% glycerol, 50 mM imidazole, 0.1 mM ATP and purity as well as concentration determined by SDS-PAGE against a BSA standard using ImageJ. For each protein preparation, the oligomerisation state and its dispersity have been checked by glycerol gradients or native PAGE. Note that for the SEC-MALS experiment, protein was eluted with 35 mM sodium phosphate buffer pH 7.0, 1 mM MgCl$_2$, 50 mM imidazole, 0.05 mM ATP and snap frozen in the absence of glycerol in 500 µl aliquots and stored in liquid nitrogen.

For purification of hTpx2 lysates were cleared and diluted 1:3 (to 50 mM NaCl) with 50 mM sodium phosphate buffer pH 7.5 as done for hKif15. Protein was allowed to bind to SP-sepharose in batch for

2 hr at 4°C. Beads were washed with a fivefold bed volume of 50 mM sodium phosphate buffer pH 7.5, 100 mM NaCl, 1 mM MgCl$_2$, 5% glycerol, 0.05% Tween-20 and protein was eluted with a threefold bed volume of 50 mM sodium phosphate buffer pH 7.5, 300 mM NaCl, 1 mM MgCl$_2$, 5% glycerol, 0.05% Tween-20, 10 mM imidazole onto Talon-beads. Protein was allowed to bind 2 hr at 4°C in batch and beads were washed with a 25-fold bed volume of 50 mM sodium phosphate buffer pH 7.5, 150 mM NaCl, 10 mM imidazole, 1 mM MgCl$_2$, 5% glycerol, 0.05% Tween-20. Protein was eluted with 50 mM sodium phosphate buffer pH 7.5, 75 mM NaCl, 150 mM imidazole, 10% glycerol. Purity and concentration was determined as above. All protein was snap frozen in 5–20 µl aliquots and stored in liquid nitrogen.

## Glycerol gradients

Glycerol gradients were essentially carried out as described in *McClelland and McAinsh (2009)* in the presence of 1 mM DTT and complete protease protein inhibitor at 4°C for 15 hr over night. We checked for hKif15/hTpx2 interactions in 50 mM HEPES pH 7.5, 1 mM MgCl$_2$, 1 mM EGTA, 0.1 mM ATP, 0/150/300/450 mM NaCl (150 mM NaCl also ± free tubulin in ATP or AMP-PNP) and 35 mM sodium phosphate buffer pH7.0, 1 mM MgCl$_2$, 1 mM EGTA, 0/150/300/500 mM NaCl, 0.1 mM ATP or AMP-PNP. The 5-mlgradient was fractionated by hand in fractions of 200 µl each. The protein was TCA-precipitated and fractions 1–23, together with $1/2$ input, analysed on the same SDS-PAGE gel stained with coomassie (SimplyBlue, Invitrogen). Protein bands were quantified with ImageJ and peak values were determined by fitting a Gaussian distribution to the data using Origin 8.5. Selected proteins from the High/low molecular weight gel filtration calibration kit (GE Healthcare) were used in standard calibration runs.

## Size-exclusion chromatography—multi angle light scattering analysis (SEC-MALS)

Protein samples were concentrated to ~2 mg/ml in 35 mM sodium phosphate buffer pH 7.0, 1 mM MgCl$_2$ and loaded onto a size exclusion column (Wyatt, Santa Barbara, CA, USA) connected to a Dawn Heleos II MALS detector and Optilab T-rEX refractometer (Wyatt). A *dn/dc* value of 0.185 was used for all calculations. Please note that for the SEC-MALS experiment only, protein-purification was modified in that hKi15 was finally eluted in 35 mM sodium phosphate buffer pH 7.0, 1 mM MgCl$_2$, 50 mM imidazole, 0.05 mM ATP and snap frozen in the absence of glycerol, which is likely to be the reason for the appearance of dimeric hKif15 in the SEC-MALS analysis that otherwise are absent in the preparation used in all the other experiments (as shown by native PAGE and glycerol gradients, see *Figure 1B,C*).

## Microtubule sedimentation assay

Microtubules for the sedimentation assay were grown in presence of 1 mM GTP in BRB80 (80 mM PIPES pH 6.8, 1 mM MgCl$_2$, 1 mM EGTA) by stepwise increasing the taxol concentration in solution. Polymerised microtubules were washed once with BRB80+Taxol using a Beckman Airfuge and resuspended in 35 mM sodium phosphate buffer pH 7.0, 1 mM MgCl$_2$, 1 mM EGTA and 1 mM DTT. Samples were mixed in the same buffer at the indicated concentrations (16.5 nM tetrameric hKif15, 33 nM dimeric hTpx2) and incubated for 20 min at room temperature in the presence of 2 mM ATP or AMP-PNP. Samples were spun for 20 min at 45k rpm in a TLA100 rotor (Beckman Coulter, High Wycombe, UK) using thick-wall polyallomer tubes (Beckman Coulter). The supernatants were TCA precipitated and analysed together with the resuspended pellets by SDS-PAGE. Protein bands were quantified using ImageJ. Since taxol stabilised microtubules are less stable in 35 mM sodium phosphate over long time periods and that hTpx2 has a stabilising effect on microtubules, we also quantified the actual amount of precipitated tubulin in order to compare both setups with and without hTpx2. This gives rise to the horizontal error bars in *Figure 5B*.

## TIRF microscopy

Preparation of sialylated coverslips and flow chamber setup (with double sided tape) was essentially carried out as described in *Bechstedt et al. (2011)*, except for the initial cleaning step with piranha solution, which had been substituted by incubation with 7.4% hydrochloric acid at 60°C over night. For single molecule assays, GMP-CPP (Jena Bioscience, Jena, Germany) stabilised, polarity-marked microtubules labelled with XRhodamin- and HiLyte-647-linked tubulin (Cytoskeleton, Denver, CO, USA) were adsorbed to the glass surface via anti beta tubulin antibodies (TUB 2.1, Sigma-Aldrich, St.Louis, MO, USA) after the flow chamber has been blocked with 1% Pluronic F-127 (Sigma-Aldrich) and 1 mg/ml kappa casein. The flow chamber was loaded with 1–10 nM His$_6$-hKIF15-GFP (tetrameric)

in BRB20 (20 mM PIPES pH6.8, 1 mM MgCl$_2$, 1 mM EGTA), 50 mM KCl, 2 mM ATP, 0.1 mg/ml kappa casein, 80 µg/ml glucose, 40 µg/ml glucose-oxidase, 16 µg/ml catalase, 1 mM DTT, sealed with VALAP (vaseline, lanolin, paraffin mixed 1:1:1) and imaged at 25°C on a Olympus CELL$^{R/TIRF}$ microscope (Olympus, Southend-on-Sea, UK) equipped with a ImagEM emCCD camera (Hamamatsu Photonics, Welwyn Garden City, UK), an environmental chamber and a stage-top-incubator (Okolab, Ottaviano, Italy) using a 100x NA 1.49 objective with 1.6x auxiliary magnification.

To visualise hKif15–eGFP movement, 3-min time-lapse videos were recorded at 2 frames per second (fps) with a 100 ms exposure using a 488 nm laser line. For the bleaching experiments, videos were recorded at 4 fps. The position of polarised microtubules before and after the time-lapse video was determined by capturing a single image with the 561 nm (100 ms exposure) and 640 nm (100 ms exposure) laser lines. In General, kymographs were generated and analysed with ImageJ and the MultipleKymograph plugin (http://www.embl.de/eamnet/html/body_kymograph.html).

For the hTpx2 inhibition experiments, equimolar dimeric hTpx2 was directly loaded together with hKif15 into the flow chamber. Imaging was started after a short incubation period of 5 min at room temperature. To prevent a strong dilution of both proteins on the microtubule lattice, a microtubule density of 150 µm tubulin per field of view (51.2 × 51.2 µm) was not exceeded.

For microtubule sliding experiments, biotinylated (Cytoskeleton) and HiLyte-647-labelled GMP-CPP-microtubules were adsorbed via streptavidin to a PEG-biotin-coated (Surface Solutions Switzerland, Dübendorf, Switzerland) cover slip that has been blocked with 1 mg/ml kappa casein. Short XRhodamin-labelled GMP-CPP-microtubules were added as cargo to the imaging-mix (as above). Again, 3 min time-lapse videos at 2 frames per second (fps) were recorded to track hKif15-GFP and cargo microtubule movement. Exposure times were 100 ms with the 488 nm laser line and 150/200 ms with the 561 nm laser. Positions of the HiLyte-647-labelled substrate microtubules were determined by capturing a single image (100 ms exposure) with a 640 nm laser line both before and after the time-lapse video.

## Laser trapping assays

For the most part, instrumentation and sample preparation was the same as described in *Carter and Cross (2005)* with the following changes: motors were non-specifically bound to polystyrene beads (550 nm, NIST calibration size standards, Polysciences, Warrington, PA, USA) by incubation in a solution of 80 mM Pipes, 2 mM MgSO$_4$, 1 mM EGTA, 1 mM DTT, 3 mg/ml D-Glucose, 0.2 mg/ml casein and 1 µM ATP at pH 7.0. Before addition to the flow cell, the incubated bead–motor solutions were diluted in an assay buffer: 20 mM Pipes, 2 mM MgSO$_4$, 1 mM ATP, 1 mM EGTA, 1 mM DTT, 3 mg/ml D-Glucose, 0.4 mg/ml casein, 4 µM taxol, and a glucose oxidase/catalase-based oxygen scavenging system. For improved stability, recordings were made in flow-cells sealed with Dow Corning high vacuum grease (Dow Corning, Barry, UK).

Motor concentration was reduced until no more than one third of beads showed any binding. A bead-binding event was defined as a bead that remained attached to the microtubule for at least 2 s after the laser trap has turned off. Each bead was tested for binding on two or more different microtubules. Bead-binding data was fitted with two poisson probability curves using a linear least-squares method (*Svoboda and Block, 1994*). This analysis showed that our experimental hKif15 and Kinesin-1 data fit a ≥1 poisson distribution (the probability that a bead binds by one or more motors), rather than the ≥2 distribution (the probability that a bead binds by two or more motors). These data confirm that we are observing single motor (i.e., tetramer) motility at the low concentrations we used in our trapping experiments (3.3 and 6.5 nM hKif15, see *Figure 2—figure supplement 2A*).

Some hKif15-linked beads bound in a diffusive non force-producing state to the microtubule, a behaviour that was rarely observed with kinesin-1-linked beads and fits to our observation of diffusive motion in our TIRF assays.

In the case of the hTpx2 flow-through recordings, the samples were not sealed at the sides to allow wash-through. A bead with active motor was recorded and a further 3–4 cell volumes of assay buffer with 18 nM dimeric hTpx2 was carefully passed through the cell whilst retaining the same bead in the trap. The trapped bead was focused close to the lower cover-glass surface during these solution changes, this offered some boundary layer protection during the solution flow and also greatly reduced the chances of a second bead entering the trap. Following recordings in the presence of hTpx2, a further 3–4 cell volumes of assay buffer were passed through the flow cell to make the final recordings following hTpx2 wash out. Bead position data were recorded at 88 kHz and averaged down to 22 kHz for analysis and storage.

## Acknowledgements

We would like to thank Miho Katsuki for providing the modified pIEx/Bac1 expression plasmid and Rob Cross for critical reading of this manuscript. We thank Warwick Medical School and Marie Curie Cancer Care for financial support.

## Additional information

### Funding

| Funder | Author |
|---|---|
| Marie Curie Cancer Care | Hauke Drechsler, Toni McHugh, Nicholas J Carter, Andrew D McAinsh |
| Warwick Medical School | Hauke Drechsler, Toni McHugh, Nicholas J Carter, Andrew D McAinsh |

The funders had no role in study design, data collection and interpretation, or the decision to submit the work for publication.

### Author contributions

HD, TMH, Conception and design, Acquisition of data, Analysis and interpretation of data, Drafting or revising the article; MRS, Acquisition of data, Analysis and interpretation of data; NJC, ADMA, Conception and design, Analysis and interpretation of data, Drafting or revising the article

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
