## [Decision Letter]

Thank you for sending your work entitled “The Kinesin-12 Kif15 is a high-force processive track-switching tetramer” for consideration at *eLife*. Your article has been favorably evaluated by a Senior editor and 2 reviewers, one of whom is a member of our Board of Reviewing Editors.

The Reviewing editor and the other reviewer discussed their comments before we reached this decision, and the Reviewing editor has assembled the following comments to help you prepare a revised submission.

Your manuscript, which describes the first in vitro experiments of the human Kinesin-12 Kif15, is an excellent contribution to the field. In general, the data are of high quality and the conclusions well supported. We would therefore like to publish your manuscript, once you have addressed a few controls that we think would strengthen the manuscript.

1) In the first paragraph of the results: what construct (His_6_-Kif15 or His_6_-Kif15-eGFP) was used to determine the molecular weight? What do the authors mean by 'native molecular weight' and 'theoretical monomeric weight' (Kif15 with/without tags)? The calculation only makes sense if one weight is given with tags and the other not... this is confusing and should be mentioned in the text. Furthermore, it is not clear how the authors calculated the molecular weight from the sedimentation coefficient and the stokes radius (a reference or an equation would be useful).

2) You observe that 14.6 % of the motors moved to the microtubule minus end. You should show a trace of a minus-end-directed run. Do they switch directions? Are the minus-end directed runs processive? What is the velocity? Could the minus-end-directed runs be diffusive rather than directed? In other words, what is the evidence that these runs are actually directed?

3) In the third paragraph of the results; 'processive particles ... bleached in up to three steps'. If the hKif15 is a tetramer why there are only three steps observable in all shown graphs (Figure 1, Figure 1–figure supplement 1E, Figure 2—figure supplement 1). It would be good to show the time on the x-axis. Furthermore, steps are hard to determine. Can the authors show a four-step bleaching trace or explain why they didn't observe one?

4) Single molecule optical tweezers experiment: You attach the hKif15 tetramer non-specifically to microspheres and used a motor-microsphere dilution where only one third showed motility. To show that one or two dimers of the tetramer walk on the microtubule poisson statistics are necessary.

5) In your experiments on the interaction between hKif15 and hTpx2 using glycerol gradient ultracentrifugation. We are worried that affinities on the order of µM may be difficult to see in this experiment if the concentrations of the proteins loaded are not high enough. Were the proteins loaded at µM concentrations? For example, in Figure 4, if one compares the intensities in lanes 13 and 18, there are slight differences which may be significant for a low affinity interaction. It is not possible to determine this with the data provided.

---

## [Author Response]

*1) In the first paragraph of the results: what construct (His*_*6*_*-Kif15 or His*_*6*_*-Kif15-eGFP) was used to determine the molecular weight? What do the authors mean by 'native molecular weight' and 'theoretical monomeric weight' (Kif15 with/without tags)? The calculation only makes sense if one weight is given with tags and the other not... this is confusing and should be mentioned in the text. Furthermore, it is not clear how the authors calculated the molecular weight from the sedimentation coefficient and the stokes radius (a reference or an equation would be useful)*.

To clarify the molecular weight of the human Kif15 motor we have carried out two new experiments. First, we have analyzed the His_6_-hKif15 protein on native gel electrophoresis. This demonstrates that hKif15 migrates at a molecular weight of approximately 730 kDa (Figure 1). Second, we analyzed the His_6_-hKif15 protein by size-exclusion chromatography/multi-angle light scattering (SEC-MALS). This is the gold standard experiment, which allows measurement of the absolute molecular mass of protein complexes. This experiment confirms that His_6_-hKif15 can exist as a tetramer with a mass of 740 kDa (Figure 1).

*2) You observe that 14.6 % of the motors moved to the microtubule minus end. You should show a trace of a minus-end-directed run. Do they switch directions? Are the minus-end directed runs processive? What is the velocity? Could the minus-end-directed runs be diffusive rather than directed? In other words, what is the evidence that these runs are actually directed*?

We now include examples of minus-end directed movement in Figure 2 – right panel. These movements were processive with a mean velocity of ∼170 nm·s^-1^. The median run length (0.8 µm verses 9.6 µm) and residency time (5.8 s versus 26.3 s) were considerably shorter than plus-end directed events and directional reversals were not observed. This new analysis is shown in Figure 2 and discussed in the text (second paragraph of the Results).

*3) In the third paragraph of the results; 'processive particles ... bleached in up to three steps'. If the hKif15 is a tetramer why there are only three steps observable in all shown graphs (*Figure 1*, Figure 1–figure supplement 1E,*
Figure 2—figure supplement 1*). It would be good to show the time on the x-axis. Furthermore, steps are hard to determine. Can the authors show a four-step bleaching trace or explain why they didn't observe one*?

We have carried out additional bleaching experiments at a higher temporal resolution (250 ms timelapse) to improve the resolution of bleaching events. Figure 2 shows example traces of hKif15-eGFP particles that bleach with three distinct steps. Note that the final GFP bleaching event to zero signal cannot be distinguished from the unbinding of the motor from the microtubule. We now make this point clear in the legend to Figure 2.

*4) Single molecule optical tweezers experiment: You attach the hKif15 tetramer non-specifically to microspheres and used a motor-microsphere dilution where only one third showed motility. To show that one or two dimers of the tetramer walk on the microtubule poisson statistics are necessary*.

We agree that Poisson statistics are important and now show these data (Figure 2—figure supplement 2) and fully describe them in the Methods section. Our experimental data fits a ≥ 1 poisson distribution (the probability that a bead binds by one or more motors), rather than the ≥ 2 distribution (the probability that a bead binds by two or more motors). These data confirm that we are observing single motor (i.e., tetramer) motility at the low concentrations we used in our trapping experiments (3.3 nM and 6.5 nM).

However, to our knowledge this type of analysis does not allow us to draw conclusions about whether one or two dimers of the tetramer attach to the microtubule. This is because of the different bead binding statistics between dimeric motors and tetrameric motors in which the two dimers are part of the same molecule and therefore inter-dependent (let us call them *virtual dimers* for the moment). The binding of the virtual dimers to the bead population is inter-dependent and therefore non-random (i.e., when a bead binds a motor it will always adsorb two virtual dimers at once). This is unlike the situation with classical dimeric motors where binding to beads would be random (i.e., dimers bind independently of each other to the beads). Given this, what is the consequence, if both virtual dimers of a tetramer were required to mediate binding of a bead to a microtubule? In this situation virtual dimers would produce a higher fraction of microtubule-bound beads when compared to the same concentration of independent dimers. In our opinion the binding behavior would not fit a ≥ 2 poisson distribution. As a result, we do not think it is possible to discriminate whether one or two dimers of the tetramer contribute to microtubule binding.

The fact that a ≥ 1 distribution is the best fit for our data does allow us to conclude that the beads have bound single tetramers (random distribution across the bead population), but we cannot exclude that both dimers of the tetramer can associate with the microtubule. In this context we would like to note that our data from the trap (like the data from TIRF) shows that the same beads can undergo different modes of motility (i.e., diffusion, processive stepping – see Figure 2). These transitions may be or may be not associated with different MT-association configurations of the tetramer (one or two dimer pairs attached). This uncertainty further precludes us from making strong conclusions as how the tetrameric hKif15 motors engage the microtubule. Future work will no doubt focus on investigating how hKif15 coordinates its four motor domains.

*5) In your experiments on the interaction between hKif15 and hTpx2 using glycerol gradient ultracentrifugation. We are worried that affinities on the order of µM may be difficult to see in this experiment if the concentrations of the proteins loaded are not high enough. Were the proteins loaded at µM concentrations? For example, in*
Figure 4*, if one compares the intensities in lanes 13 and 18, there are slight differences which may be significant for a low affinity interaction. It is not possible to determine this with the data provided*.

This is an excellent point. We have repeated the glycerol gradient experiments using 2.5 µM dimeric hTpx2 and 80nM tetrameric hKif15. Under these conditions we can now detect the formation of the hKif15-hTpx2 complex in solution. This data is now shown in Figure 5. However, we note that the affinity of this interaction is low and our data are consistent with our original conclusion that the stable Kif15-hTpx2 complex formation involves binding to the microtubule track: in the presence of microtubules, low nM concentrations hTpx2 increase the microtubule-binding affinity of hKif15 and inhibit the motors motility (see Figure 5).